# AutoQ: Automated Kernel-Wise Neural Network Quantization[*]

Qian Lou, Feng Guo, Minje Kim, Lantao Liu, and Lei Jiang

{louqian, fengguo, minje, lantao, jiang60}@iu.edu
Indiana University Bloomington

## Abstract

Network quantization is one of the most hardware friendly techniques to enable the deployment of convolutional neural networks (CNNs) on low-power mobile devices. Recent network quantization techniques quantize each weight kernel in a convolutional layer independently for higher inference accuracy, since the weight kernels in a layer exhibit different variances and hence have different amounts of redundancy. The quantization bitwidth or bit number (QBN) directly decides the inference accuracy, latency, energy and hardware overhead. To effectively reduce the redundancy and accelerate CNN inferences, various weight kernels should be quantized with different QBNs. However, prior works use only one QBN to quantize each convolutional layer or the entire CNN, because the design space of searching a QBN for each weight kernel is too large. The hand-crafted heuristic of the kernel-wise QBN search is so sophisticated that domain experts can obtain only sub-optimal results. It is difficult for even deep reinforcement learning (DRL) Deep Deterministic Policy Gradient (DDPG)-based agents to find a kernel-wise QBN configuration that can achieve reasonable inference accuracy. In this paper, we propose a hierarchical-DRL-based kernel-wise network quantization technique, AutoQ, to automatically search a QBN for each weight kernel, and choose another QBN for each activation layer. Compared to the models quantized by the state-of-the-art DRL-based schemes, the same models quantized by AutoQ reduce the inference latency by 54.06%, and decrease the inference energy consumption by 50.69% averagely, while achieving the same inference accuracy.

## 1 Introduction

Although convolutional neural networks (CNNs) have been the dominant approach (Sandler et al., 2018) to solving a wide variety of problems such as computer vision and recommendation systems, it is challenging to deploy CNNs to mobile devices having only limited hardware resources and tight power budgets, due to their huge essential computing overhead, e.g., an inference of MobileNetV2 (Sandler et al., 2018) involves $6.9M$ weights and $585M$ floating point operations.

Several approaches such as pruning (He et al., 2018) and low-rank approximation (Denton et al., 2014) are proposed to reduce the inference computing overhead of CNNs. Network quantization (Wang et al., 2019; Lin et al., 2017) becomes one of the most hardware friendly CNN acceleration techniques by approximating real-valued weights and activations to $QBN$-bit fixed-point representations, and performing inferences using cheaper fixed-point multiple-accumulation (MAC) operations, where $QBN$ is the *quantization bit number*.

Instead of using one QBN for the whole CNN, the *layer-wise* network quantization (Wang et al., 2019; Elthakeb et al., 2018) assigns a QBN to the weights of each convolutional layer, and searches another QBN for the activations of the same layer to decrease the inference computing overhead. But the inference cost of the layer-wise quantized CNNs is still prohibitive for low-power mobile devices powered by batteries. Recent works (Zeng et al., 2019; Choukroun et al., 2019b; Zhang et al., 2018; Li et al., 2019; Krishnamoorthi, 2018; Sasaki et al., 2019) find that various weight kernels of a

---

[*]This work was supported in part by NSF CCF-1908992 and CCF-1909509.

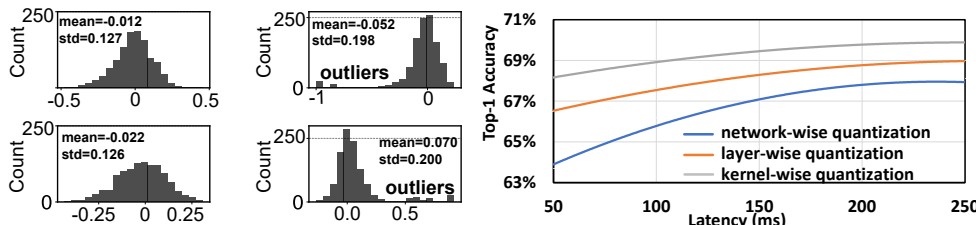

Figure 1: The weight distribution of kernels.  Figure 2: Inference accuracy and latency.

convolutional layer (ResNet-18) exhibit different variances shown in Figure 1 and hence have different amounts of redundancy. Therefore, they quantize each weight kernel independently for higher accuracy by calculating a $QBN$-element scaling factor vector for each kernel, rather than globally quantize all the kernels of a layer as a whole. To reduce different amounts of redundancy among different weight kernels, these *kernel-wise* network quantization techniques should have searched a QBN for each kernel of each layer in a CNN. However, the search space of choosing a QBN for each weight kernel is too large, so prior kernel-wise network quantization (Zeng et al., 2019; Choukroun et al., 2019b; Zhang et al., 2018; Li et al., 2019; Krishnamoorthi, 2018; Sasaki et al., 2019) still uses the same QBN for the entire CNN. As Figure 2 shows, compared to the layer-wise quantized model, on the same FPGA accelerator (Umuroglu et al., 2019a), the kernel-wise quantized model (assigning a QBN to each weight kernel and choosing a QBN for each activation layer) improves the inference accuracy by $\sim 2\%$ (ImageNet) with the same computing overhead (inference latency).

How to decide a QBN for each weight kernel is the most important task of the kernel-wise network quantization, since the QBNs have a large impact on the inference accuracy, latency and hardware overhead. Determining a QBN for each weight kernel via hand-crafted heuristics is so sophisticated that even machine learning experts can obtain only sub-optimal results. Recent works (Wang et al., 2019; Elthakeb et al., 2018) automatically select a QBN for each layer of a CNN through a deep reinforcement learning (DRL) agent without human intervention. However, it is still difficult for low-power mobile devices such as drones and smart glasses to adopt the layer-wise quantized CNN models. These mobile devices are very sensitive to the bit-width of fixed-point MAC operations and memory access during inferences due to their limited battery lifetime and hardware resources. Kernel-wise network quantization assigning a QBN to each weight kernel and searching a QBN for each activation layer of a CNN becomes a must to enable the efficient deployment of deep CNNs on mobile devices by reducing the inference computing overhead. Although it is straightforward to perform kernel-wise quantization via DRL, it takes ultra-long time for a DRL agent to find a proper QBN for each weight kernel of a CNN. As CNN architectures are becoming deeper, it is infeasible to employ rule-based domain expertise or conventional DRL-based techniques to explore the exponentially enlarging search space of kernel-wise network quantization.

In this paper, we propose a hierarchical-DRL-based agent, *AutoQ*, to automatically and rapidly search a QBN for each weight kernel and choose a QBN for each activation layer of a CNN for accurate kernel-wise network quantization. AutoQ comprises a high-level controller (HLC) and a low-level controller (LLC). The HLC chooses a QBN for each activation layer and generates a goal, the average QBN for all weight kernels of a convolutional layer, for each layer. Based on the goal, the LLC produces an action, QBN, to quantize each weight kernel of the layer. The HLC and LLC simultaneously learn by trials and errors, i.e., penalizing inference accuracy loss while rewarding a smaller QBN. We also build a state space, a goal and an action space, an intrinsic reward and an extrinsic reward for AutoQ. Instead of proxy signals including FLOPs, number of memory access and model sizes, we design the extrinsic reward to take the inference latency, energy consumption and hardware cost into consideration.

## 2  BACKGROUND AND RELATED WORK

**Quantization**. Recent works (Lin et al., 2016; Zhou et al., 2017; Jacob et al., 2018; McKinstry et al., 2018; Zhang et al., 2018) quantize the real-valued weights and activations to fixed-point representations, so that the model size is reduced and inferences can use low-cost fixed-point MAC operations. To further reduce inference computing overhead, prior works (Kim & Smaragdis, 2016; Xu et al., 2018; Guo et al., 2017; Tang et al., 2017; Rastegari et al., 2016; Lin et al., 2017) quantize weights and activations into multi-bit binary codes of {-1, +1}s. Rather than real-valued MACs, inferences of these quantized models depend on bit-wise logic operations, i.e., XNORs and popcounts. These

traditional quantization techniques either simply assign a single QBN to the whole CNN or require domain experts to determine a QBN for each layer of a CNN.

Table 1: The search space size of network quantization. $QBN \in [0, 32]$, where 0 means the component is pruned. $n_{layer}$ is the layer number of the network.

| quantization granularity | search space size (weight $\times$ activation) |
|---|---|
| network-wise | $33 \times 33$ |
| layer-wise | $33^{n_{layer}} \times 33^{n_{layer}}$ |
| kernel-wise | $33^{\sum_{i=1}^{n_{layer}} c_{outi}} \times 33^{n_{layer}}$ |

**Kernel-wise quantization**. As Table 1 shows, almost all prior works (Lin et al., 2016; Kim & Smaragdis, 2016; Rastegari et al., 2016; Lin et al., 2017; Guo et al., 2017; Zhou et al., 2017; Jacob et al., 2018; Tang et al., 2017; Xu et al., 2018; McKinstry et al., 2018; Zhang et al., 2018) categorized as the *network-wise* quantization focus on searching a $QBN \in [0, 32]$ for all weights, and searching another QBN for all activations in a CNN. Totally, there are only 1089 combinations of the QBN configuration for the network-wise quantization. The layer-wise quantization (Wang et al., 2019) searches a $QBN \in [0, 32]$ for all weights of a convolutional layer, and decides another QBN for all activations of the same layer. The QBN search space size of the layer-wise quantization substantially increases to $33^{n_{layer}} \times 33^{n_{layer}}$, where $n_{layer}$ is the layer number of a CNN. Recent works (Zeng et al., 2019; Choukroun et al., 2019b; Zhang et al., 2018; Li et al., 2019; Krishnamoorthi, 2018; Sasaki et al., 2019) observe various weight kernels of a convolutional layer have different amounts of redundancy, and quantize each weight kernel independently for higher accuracy. To exploit different amounts of redundancy among different weight kernels, these kernel-wise network quantization techniques should have searched a QBN for each kernel of each convolutional layer, and assigned a QBN for each activation layer in a CNN. However, the search space size of the kernel-wise network quantization is $33^{\sum_{i=1}^{n_{layer}} c_{outi}} \times 33^{n_{layer}}$, where $c_{outi}$ is the number of weight kernels (output channels) of the $i$th layer. No prior work tries to search such huge design space.

Table 2: The comparison of DRL-based techniques for quantization and pruning.

| feature | AMC | ReLeQ | HAQ | AutoQ |
|---|---|---|---|---|
| search for activations and weights | ✗ | ✗ | ✓ | ✓ |
| kernel-wise quantization | ✗ | ✗ | ✗ | ✓ |
| hierarchical DRL | ✗ | ✗ | ✗ | ✓ |
| shaped intrinsic reward | ✗ | ✗ | ✗ | ✓ |

**AutoML**. Recent works take advantage of DRL (Baker et al., 2016; Zoph et al., 2017), genetic algorithm (Suganuma et al., 2017; Stanley & Miikkulainen, 2002) and Bayesian Optimization (Kandasamy et al., 2018; Stewart & Stalzer, 2018) to automatically architect CNNs for higher inference accuracy. Their network architectures outperform many human-designed neural networks. The weight channel pruning is automatically conducted by DRL (He et al., 2018) and genetic algorithm (Wang et al., 2018). ReLeQ (Elthakeb et al., 2018) quantizes only the weights of each layer of a CNN by DRL, while HAQ (Wang et al., 2019) performs the layer-wise quantization for both weights and activations via a DRL agent. No prior quantization or pruning work relies on hierarchical DRL. Table 2 compares AutoQ against prior DRL-based techniques for quantization and pruning. AutoQ is the **first** work to automatically quantize each weight kernel and each activation layer of a pre-trained CNN model for mobile devices by hierarchical DRL.

# 3   AUTOQ

**Overview**. We do not aim to present a new network quantization technique, but we formulate the search of a QBN for each weight kernel and each activation layer as a hierarchical DRL problem. We propose a two-level hierarchical DRL technique, AutoQ, to automatically quantize the weights in the kernel-wise manner and the activations in the layer-wise fashion. We build the state space, action and goal space, extrinsic and intrinsic reward functions and a hierarchical DRL agent for AutoQ. Although we use the state-of-the-art learned quantization technique, LQ-Nets (Zhang et al., 2018), to quantize weight kernels and activation layers with the QBNs found by AutoQ, future novel quantization techniques can be easily integrated to AutoQ to improve the inference accuracy of the quantized networks. In the extrinsic reward, besides the inference latency and energy (Wang et al., 2019), AutoQ also considers the FPGA area overhead critical to low-cost mobile devices.

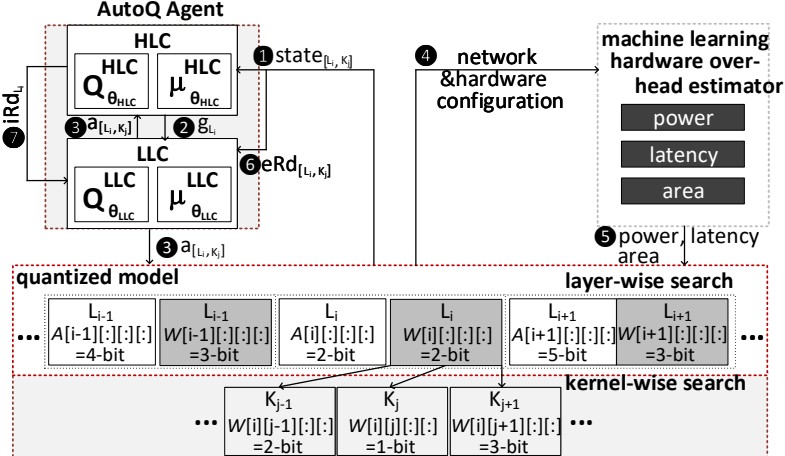

Figure 3: The working flow of AutoQ (HLC: high-level controller, LLC: low-level controller).

**Working Flow**. For an $n_{layer}$-layer CNN, the weight is defined as $\mathbf{W} \in \mathbb{R}^{n_{layer} \times c_{out} \times c_{in} \times w_w \times h_w}$, where $n_{layer}$ is the number of layers; $c_{out}$ denotes the number of kernels (output channels); $c_{in}$ means the number of input channels; $w_w$ indicates the kernel width; and $h_w$ is the kernel height. The activation is defined as $\mathbf{A} \in \mathbb{R}^{n_{layer} \times c_{in} \times w_a \times h_a}$, where $w_a$ is the feature map width; and $h_a$ means the feature map height. The working flow of AutoQ is shown in Figure 3. AutoQ consists of a high-level controller (HLC) and a low-level controller (LLC). The HLC quantizes the network layer by layer, while the LLC searches a QBN for each weight kernel in a layer. ❶ At first, AutoQ receives an observation $state_{[L_i, K_j]}$ from the environment that is the quantized network model, where $state_{[L_i, K_j]}$ includes the information of the CNN architecture. ❷ The HLC makes a goal $g_{L_i}$ that is the QBN for the activation layer $L_i$. The flow then jumps to ❹. Or the HLC generates a goal $g_{L_i}$ which is the average QBN of all weight kernels in the layer $L_i$ for the LLC. ❸ The LLC produces an action $a_{[L_i, K_j]}$, QBN, for the weight kernel $K_j$ of the layer $L_i$. For the entire layer $L_i$, the LLC aims to reach the goal $g_{L_i}$ of the HLC. ❹ The environment sends the network quantization and hardware configuration to the fast and accuracy machine-learning-based hardware overhead estimator. ❺ The hardware overhead estimator returns the energy consumption, area overhead and inference latency for the current quantization and hardware configuration. ❻ With the hardware overhead and inference accuracy, the environment generates an extrinsic reward $eRd_{[L_i, K_j]}$ for AutoQ to evaluate the LLC action. ❼ Based on all actions of LLC for the layer $L_i$, the HLC provides an intrinsic reward $iRd_{L_i}$ to tell how well the goal is implemented by the LLC.

**State Space**. A state $state_{[L_i, K_j]}$ (observation) is represented by

$$state_{[L_i, K_j]} = (L_i, K_j, c_{in}, c_{out}, s_{kernel}, s_{stride}, s_{feature}, b_{dw}, b_{w/a}, g_{L_{i-1}}, a_{[L_i, K_{j-1}]}) \qquad (1)$$

where $L_i$ is the layer index; $K_j$ means the weight kernel index; $c_{in}$ indicates the number of input channels; $c_{out}$ denotes the number of kernels; $s_{kernel}$ is the kernel size; $s_{stride}$ is the stride; $s_{feature}$ is the input feature map size; $b_{dw}$ binarily indicates depthwise convolution or not; $b_{w/a}$ binarily represents weight or activation; $g_{L_{i-1}}$ is the goal (average QBN) of the last layer; and $a_{[L_i, K_{j-1}]}$ is the action (QBN) of the last kernel in the $L_i$ layer. For each variable in $state_{[L_i, K_j]}$, we normalize it to $[0, 1]$. If the layer is a fully-connected layer, we set $s_{kernel} = 1$, $s_{stride} = 0$, and $b_{dw} = 0$.

**Goal and Action Space**. The HLC produces the average QBN for all weight kernels of each layer or the QBN for each activation layer as a goal, while the LLC generates a QBN for each weight kernel in a layer as an action. The HLC goal $g_{L_i}$ for the $L_i$ layer uses a *continuous* space and can be any real value between 1 and $goal_{max}$, where $goal_{max}$ is the maximum average QBN for a layer and we set it to 8. If the $L_i$ layer is an activation layer, we round the real-valued $g_{L_i}$ to the discrete value of $roundup(1 + g_{L_i} \cdot (goal_{max} - 1))$. Although the LLC action is an integer between 0 and $action_{max}$, it still uses a continuous space to capture the relative order, i.e., 2-bit is more aggressive than 3-bit, where $action_{max}$ is the maximum QBN for a kernel and we set it to 8. For the $K_j$ kernel of the $L_i$ layer, the LLC generates the continuous action $ra_{[L_i, K_j]}$ that is in the range of $[0, 1]$, and round it up to the discrete value $a_{[L_i, K_j]} = roundup(ra_{[L_i, K_j]} \cdot action_{max})$.

**Extrinsic Reward**. After an action $a_{[L_i,K_j]}$ is taken, AutoQ arrives at a new state $state_{[L_i,K_{j+1}]}$ and receives an extrinsic reward $eRd$ from the environment. The HLC aims to maximize the accumulative extrinsic reward $eRd = \sum_i \sum_j \gamma_{eRd}^{\sum_i c_{outi}+j-1} eRd_{[L_i,K_j]}$, where $\gamma_{eRd} \in [0,1)$ is a decay factor. The immediate extrinsic reward can be represented by

$$eRd_{[L_i,K_j]}(NC,HC) = log(\frac{accuracy(NC)^{\psi_{acc}}}{lat(NC,HC)^{\psi_l} \cdot en(NC,HC)^{\psi_e} \cdot area(NC,HC)^{\psi_a}}) \quad (2)$$

where $NC$ is the network configuration; $HC$ means the hardware configuration, e.g., memory bandwidth; $accuracy(NC)$ indicates the inference accuracy; $lat$ is the inference latency of the network $NC$ running on the hardware $HC$; $en$ represents the inference energy of $NC$ running on $HC$; $area$ is the FPGA area (hardware cost) used by $NC$ on $HC$; $\psi_{acc}$, $\psi_l$, $\psi_e$ and $\psi_a$ are user-defined factors deciding the impact of inference accuracy, latency, energy and FPGA area on the extrinsic reward. By different values of user-defined factors, AutoQ implements the *resource-constrained* and *accuracy-guaranteed* searches. For resource-constrained applications, e.g., low-power drones, AutoQ sets $\psi_{acc} = 1$, $\psi_l = 0$, $\psi_e = 0$ and $\psi_a = 0$ to achieve the best accuracy given the maximum amount of hardware resources (latency, energy, and FPGA area). This extrinsic reward offers no incentive for lower QBNs, so AutoQ reduces the QBN by limiting the action space. AutoQ allows arbitrary action at the first few layers and starts to limit the action when it finds that the hardware resource budget is insufficient even after using the smallest QBN for all the following layers. For accuracy-guaranteed applications, e.g., fingerprint locks, AutoQB sets $\psi_{acc} = 2$, $\psi_l < 1$, $\psi_e < 1$ and $\psi_a < 1$ to obtain the shortest latency, the minimal energy, and the smallest hardware cost with no accuracy loss.

**Intrinsic Reward**. Based on the goal $g_{L_i}$ produced by the HLC for the $L_i$ layer, the LLC generates $c_{out}$ actions $a_{[L_i,K_0]} \sim a_{[L_i,K_{c_{out}-1}]}$ at the states $state_{[L_i,K_0]} \sim state_{[L_i,K_{c_{out}-1}]}$. AutoQ then arrives the state $state_{[L_i,K_{c_{out}-1}]}$, where it receives an intrinsic reward $iRd$ and maximizes the accumulative intrinsic reward $iRd = \sum_j \gamma_{iRd}^{j-1} iRd_{[L_i,K_j]}$, where $\gamma_{iRd} \in [0,1)$ is a decay factor and $iRd_{[L_i,K_j]}$ indicates the intrinsic reward for the weight kernel $K_i$ of the layer $L_i$. The LLC produces actions to help the HLC to maximize the extrinsic reward, so it should aim to complete the goal of the HLC and to maximize the extrinsic reward. But at the beginning of the AutoQ training, the extremely low extrinsic reward due to the random goals of the HLC prevents the LLC from efficiently learning from the environment. We propose a shaped reward as the intrinsic reward for the LLC to take both the goal completion and the extrinsic reward into consideration, and to enable fine-grained low-level behavior learning. The intrinsic reward can be represented by

$$iRd_{L_i} = (1-\zeta) \cdot (-||g_{L_i} \cdot c_{out} - \sum_{j=0}^{c_{out}-1} a_{L_i,K_j}||_2) + \zeta \cdot \sum_{j=0}^{c_{out}-1} eRd_{L_i,K_j} \quad (3)$$

where $\zeta$ is a user-defined factor dynamically enlarging from 0.1 to 0.8 as the number of training epochs increases. When $\zeta$ is small, the HLC has stronger influence on the LLC. On the contrary, when $\zeta = 1$, the LLC maximizes only the accumulative extrinsic reward.

**Hardware Overhead Estimator**. A recent work (Wang et al., 2019) estimates the hardware latency and energy by physical FPGA accelerators. However, a typical synthesis for a CNN model on a FPGA costs $> 30$ minutes (Gopinath et al., 2019). Invoking a FPGA synthesis for each action will make AutoQ unacceptably slow. We adopt fast and accurate FPGA latency, area (Liu & Carloni, 2013) and power (Zhou et al., 2019) models to predict the inference latency, energy and FPGA area for an arbitrary configuration of network and hardware. These machine-learning-based models are highly accurate and can estimate the hardware overhead to compute the extrinsic reward of AutoQ within several milliseconds.

**Hierarchical DRL**. AutoQ uses a HIerarchical Reinforcement learning with Off-policy correction (HIRO) (Nachum et al., 2018), to implement the HLC and the LLC. The LLC is trained by incorporating $g_{L_i}$ into the standard TD3 method (Nachum et al., 2018). So the low-level Q-value function $Q_{\theta_{LLC}}^{LLC}$ is to minimize the error $\varepsilon_{LLC}(state_{[L_i,K_j]}, g_{L_i}, a_{[L_i,K_j]}, state_{[L_i,K_{j+1}]})$, which is

$$(Q_{\theta_{LLC}}^{LLC}(state_{[L_i,K_j]}, g_{L_i}, a_{[L_i,K_j]}) - iRd_{L_i} - \gamma_{iRd} \cdot Q_{\theta_{LLC}}^{LLC}(state_{[L_i,K_{j+1}]}, g_{L_i}, \mu_{\phi_{LLC}}^{LLC}(state_{[L_i,K_{j+1}]}, g_{L_i})))^2$$

$$(4)$$

where $\mu_{\phi_{LLC}}^{LLC}$ is trained to maximize $Q_{\theta_{LLC}}^{LLC}$. We further augment $\mu_{\phi_{LLC}}^{LLC}$ with Gaussian noises by collecting the actions as $N(\mu_{\phi_{LLC}}^{LLC}, \sigma_{a_{[L_i,K_j]}})$, where $N$ is a Gaussian distribution, and $\sigma_{a_{[L_i,K_j]}}$ is

the variance. During the exploitation, $\sigma_{a_{[L_i,K_j]}}$ is initialized to 0.5 and decayed after each episode exponentially. The HLC converts a series of high-level transition tuples

$$(s_{[L_i,K_0:K_{c_{out}-1}]}, g_{L_i}, a_{[L_i,K_0:K_{c_{out}-1}]}, eRd_{[L_i,K_0:K_{c_{out}-1}]}, s_{[L_{i+1},K_0]}) \tag{5}$$

to state-goal-reward transitions

$$(s_{[L_i,K_0]}, g_{L_i}, \sum eRd_{[L_i,K_0:K_{c_{out}-1}]}, s_{[L_{i+1},K_0]}) \tag{6}$$

where $a_{[L_i,K_0:K_{c_{out}-1}]}$ denotes the sequence of $a_{[L_i,K_0]} \sim a_{[L_i,K_{c_{out}-1}]}$; and $eRd_{[L_i,K_0:K_{c_{out}-1}]}$ means the sequence of $eRd_{[L_i,K_0]} \sim eRd_{[L_i,K_{c_{out}-1}]}$. AutoQ stores these state-goal-reward transitions into the replay buffer. However, since transitions obtained from the past LLCs do not accurately reflect the actions that would occur if the same goal was used with the current LLC, AutoQ has to introduce a correction translating old transitions into ones that agree with the current LLC. AutoQ re-labels the high-level transition $(s_{[L_i,K_0]}, g_{L_i}, \sum eRd_{[L_i,K_0:K_{c_{out}-1}]}, s_{[L_{i+1},K_0]})$ with a different goal $\tilde{g_{L_i}}$ chosen to maximize the probability $\mu_{\phi_{LLC}}^{LLC}(a_{[L_i,K_0:K_{c_{out}-1}]}|s_{[L_i,K_0:K_{c_{out}-1}]}, \tilde{g_{L_i}})$. AutoQ computes 10 candidate goals sampled randomly from a Gaussian distribution centered at $g_{L_i}$, and selects the minimal goal to re-label the experience.

**Quantization and Finetuning**. During a search, we quantize the model by the learned quantization technique (Zhang et al., 2018), and finetune the quantized model for ten epochs to recover the accuracy using stochastic gradient descent (SGD) with a fixed learning rate of $10^{-3}$ and momentum of 0.9. We randomly select 100 categories from the ImageNet to accelerate the model finetuning. After the search is done, we quantize the model with the best policy found by AutoQ and finetune it on the full dataset.

**Implementation Details**. An AutoQ agent, i.e., HLC or LLC, consists of an actor network and a critic network. Both share the same architecture, i.e., two hidden layers, each of which has 300 units. For the actor network, we add an additional sigmoid function producing an output in the range of $[0, 1]$. We use a fixed learning rate of $10^{-4}$ for the actor network and $10^{-3}$ for the critic network. AutoQ trains the networks with the batch size of 64 and the replay buffer size of 2000. AutoQ first explores 100 episodes with a constant noise, i.e., $\delta_{a_{[L_i,K_j]}} = 0.5$ for the LLC and $\delta_{g_{[L_i]}} = 0.5$ for the HLC, and then exploits 300 episodes with exponentially decayed noise.

**Storage Cost**. We need to record a 4-bit QBN ranging from 0 to 8 for each activation layer and each weight kernel of a convolutional layer. The storage overhead of AutoQ is $\sim 0.1\%$ of the size of various CNN models. For instance, ResNet-18 found by resource-constrained AutoQ requires 8.3MB to store its quantized model in Table 3. The storage overhead of AutoQ is only 0.07%.

## 4 EXPERIMENTAL RESULTS

**Experimental Settings**. To evaluate AutoQ, we selected several CNN models including ResNet-18, ResNet-50, SqueezeNetV1 (Iandola et al., 2016) and MobileNetV2 (Sandler et al., 2018). The CNN models are trained on ImageNet including 1.26M training images and tested on 50K test images spanning 1K categories of objects. We evaluated the inference performance, energy consumption and FPGA area of the CNN models quantized by AutoQ on a Xilinx Zynq-7020 embedded FPGA. On the FPGA, we implemented a temporal CNN accelerator (Umuroglu et al., 2019b) that uses bit-serial multipliers, each of which computes with one-bit digits from multiple weights and their corresponding activations in parallel at one time, and then accumulates their partial products.

### 4.1 OVERALL PERFORMANCE

**Resource-constrained Quantization**. We make AutoQ perform the resource-constrained searches by imposing a latency constraint and setting $\psi_{acc} = 1$, $\psi_l = 0$, $\psi_e = 0$ and $\psi_a = 0$ in the extrinsic reward. With such a setting, AutoQ aims to search for the best inference accuracy given the longest latency constraint, which is set to the inference latency of the 4-bit network-wise quantized CNN models. We compare the kernel-wise AutoQ quantized models against the layer-wise Hardware-Aware Automated Quantization (HAQ) (Wang et al., 2019) quantized models and the 4-bit network-wise quantized models in Table 3. We used the LQ-Nets quantization (Zhang et al., 2018) to quantize and finetune the models in all three schemes. The network-wise scheme uses 4-bit to quantize the whole models, while the layer-wise scheme searches a QBN for weights of each layer, and chooses another QBN for activations of the same layer. AutoQ chooses a QBN for each weight kernel, and selects another QBN for each activation layer of a CNN. In Table 3, the average QBN of weights

Table 3: Network Quantization by AutoQ (A-QBN: the average QBN of activations; W-QBN: the average QBN of weights; LAT: inference latency).

| model | scheme | resource-constrained | | | | | accuracy-guaranteed | | | | |
|---|---|---|---|---|---|---|---|---|---|---|---|
| | | top-1 err (%) | top-5 err(%) | A-QBN (bit) | W-QBN (bit) | LAT (ms) | top-1 err (%) | top-5 err(%) | A-QBN (bit) | W-QBN (bit) | LAT (ms) |
| ResNet-18 | network-wise | 32.7 | 12.32 | 4 | 4 | 296.8 | 32.7 | 12.32 | 4 | 4 | 296.8 |
| | layer-wise | 31.8 | 11.92 | 3.32 | 4.63 | 290.9 | 32.5 | 11.90 | 3.37 | 3.65 | 189.6 |
| | kernel-wise | **30.22** | **11.62** | 4.12 | 3.32 | 286.3 | 32.6 | 11.82 | **3.02** | **2.19** | 125.3 |
| | original | 30.10 | 11.62 | 16 | 16 | 1163 | 30.10 | 11.62 | 16 | 16 | 1163 |
| ResNet-50 | network-wise | 27.57 | 9.02 | 4 | 4 | 616.3 | 27.57 | 9.02 | 4 | 4 | 616.3 |
| | layer-wise | 26.79 | 8.32 | 4.23 | 3.51 | 612.3 | 27.49 | 9.15 | 4.02 | 3.12 | 486.4 |
| | kernel-wise | **25.53** | **7.92** | 3.93 | 4.02 | 610.3 | 27.53 | 9.12 | **3.07** | **2.21** | 327.3 |
| | original | 25.20 | 7.82 | 16 | 16 | 2357 | 25.20 | 7.82 | 16 | 16 | 2357 |
| SqueezeNetV1 | network-wise | 45.67 | 23.12 | 4 | 4 | 43.1 | 45.67 | 23.12 | 4 | 4 | 43.1 |
| | layer-wise | 44.89 | 21.14 | 3.56 | 4.27 | 42.1 | 45.63 | 23.04 | 3.95 | 3.28 | 25.5 |
| | kernel-wise | **43.51** | **20.89** | 4.05 | 3.76 | 41.6 | 45.34 | 23.02 | **3.29** | **2.32** | 12.5 |
| | original | 43.10 | 20.5 | 16 | 16 | 127.3 | 43.10 | 20.5 | 16 | 16 | 127.3 |
| MobileNetV2 | network-wise | 31.75 | 11.67 | 4 | 4 | 37.4 | 31.35 | 11.67 | 4 | 4 | 37.4 |
| | layer-wise | 30.98 | 10.57 | 3.57 | 4.22 | 36.9 | 31.34 | 10.57 | 3.92 | 3.21 | 23.9 |
| | kernel-wise | **29.20** | **9.67** | 4.14 | 3.67 | 36.1 | 31.32 | 11.32 | **3.13** | **2.26** | 10.2 |
| | original | 28.90 | 9.37 | 16 | 16 | 123.6 | 28.90 | 9.37 | 16 | 16 | 123.6 |

(W-QBN) can be calculated by

$$\frac{\sum_{L_i=1}^{n_{layer}} \sum_{K_j=1}^{c_{couti}} Weight\_QBN_{[L_i,K_j]}}{\sum_{i=1}^{n_{layer}} c_{couti}} \tag{7}$$

where $c_{outi}$ is the number of output channels in the layer $L_i$ and $Weight\_QBN_{[L_i,K_j]}$ is the QBN for the $K_j$th weight kernel in the layer $L_i$. The average QBN of activations (A-QBN) is computed as $\frac{\sum_{L_i=1}^{n_{layer}} Act\_QBN_{L_i}}{n_{layer}}$, where $Act\_QBN_{L_i}$ is the QBN for all activations of the layer $L_i$. Compared to the layer-wise quantization, AutoQ improves the top-1 inference accuracy by $> 1.25\%$ when spending almost the same inference latency. Compared to the 16-bit full-precision models, the models quantized by AutoQ degrade the inference accuracy by at most only $0.41\%$, but reduce the inference latency by $71.2\%$ on average.

**Accuracy-guaranteed Quantization**. We run AutoQ to do the accuracy-guaranteed searches by setting $\psi_{acc} = 2$, $\psi_l = 0.5$, $\psi_e = 0$ and $\psi_a = 0$ in the extrinsic reward. Such an extrinsic reward drives AutoQ to quantize the models to achieve the shortest inference latency without significant accuracy loss. Compared to the layer-wise scheme, AutoQ substantially reduces the inference latency by 42.2% while achieving a similar (averagely -0.1%) top-1 inference accuracy. Compared to ResNet-18 and ResNet50, the compact models such as SqueezeNetV1 suffer from larger top-1 accuracy degradation, i.e., -0.3% in a accuracy-guaranteed search of AutoQ.

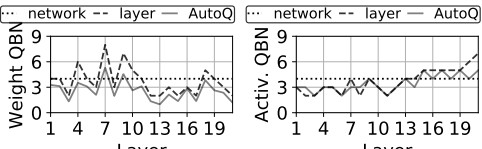
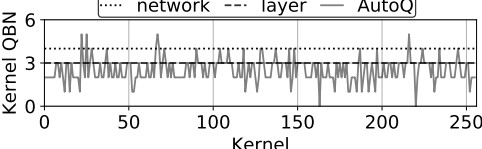

Figure 4: The ave. QBNs in various layers. Figure 5: The weight kernel QBNs in a layer.

## 4.2 DETAILED ANALYSIS

**Kernel-wise Search**. AutoQ can assign a QBN to each kernel of a convolutional layer. The average weight QBN and the average activation QBN of each ResNet-18 layer found by an accuracy-guaranteed AutoQ search are shown in Figure 4. Both the network-wise and layer-wise quantization techniques use only one QBN to quantize all weight kernels in a convolutional layer, and quantize all activations of the layer by another QBN. On the contrary, AutoQ searches a QBN for each weight kernel. Compared to a CNN model quantized by the network-wise or layer-wise quantization technique, the same model quantized by the kernel-wise AutoQ can achieve similar inference accuracy but with a smaller average QBN in each layer. We also show the weight kernel QBNs of the $L_{14}$ layer of ResNet-18 produced by resource-constrained AutoQ searches in Figure 5. AutoQ automatically identifies which weight kernel has a smaller (larger) variance and thus less (more) redundancy, so that it can assign a larger (smaller) QBN to the weight kernel. For instance, as Figure 1 shows,

compared to the 53th weight kernel (top-right), the 52th weight kernel (top-left) of ResNet-18 has a smaller weight distribution variance. Therefore, in Figure 5, AutoQ assigns a smaller QBN to the 52th weight kernel but provides the 53th weight kernel a larger QBN.

**Hierarchical DRL Agent with Shaped Intrinsic Reward**. We evaluated and compared our hierarchical-DRL-based AutoQ against the traditional one-level DDPG-based DRL adopted by a recent layer-wise quantization technique, HAQ (Wang et al., 2019). The reward comparison of different techniques during the kernel-wise quantization on MobileNetV2

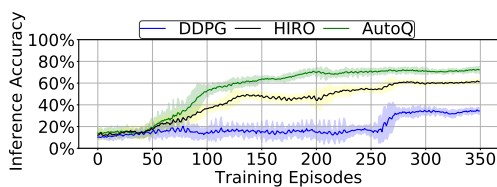

Figure 6: The DRL scheme comparison.

is shown in Figure 6. HAQ and AutoQ both support resource-constrained searches, but HAQ cannot support accuracy-guaranteed searches. So their rewards are just the inference accuracy. Through the goals of the HLC and the actions of the LLC, AutoQ can find a QBN for each weight kernel and achieve $> 70\%$ accuracy much faster than the DDPG-based DRL, i.e., it reaches $\sim 70\%$ accuracy after only 200 episodes. However, the DDPG-based DRL is stuck with $20\%$ inference accuracy until 250 episodes. The hierarchical-DRL-based AutoQ significantly accelerates the search space exploration of the kernel-wise network quantization. Although AutoQ uses a prior hierarchical DRL agent HIRO (Nachum et al., 2018) to search a QBN for each weight kernel, we propose a novel shaped intrinsic reward considering both the completion of the HLC goals and the extrinsic reward to accelerate the search. The intrinsic reward of HIRO takes only the completion of the HLC goals into consideration. The LLC of HIRO cannot directly learn from the environment. Therefore, compared to AutoQ, it takes extra 200 episodes for HIRO to reach only $60\%$ accuracy as shown in Figure 6.

**Extrinsic Reward**. Unlike the reward of the DDPG-based layer-wise HAQ (Wang et al., 2019) considering only the inference accuracy, the extrinsic reward of AutoQ can balance the trade-off between the inference accuracy, latency, energy consumption and FPGA area by enabling the accuracy-guaranteed search. By setting $\psi_{acc} = 2$, $\psi_l = 0.5$, $\psi_e = 0.5$ and $\psi_a = 0.5$, AutoQ takes the inference accuracy, latency, energy and FPGA area into consideration during an accuracy-guaranteed search. For instance, AutoQ can find two kernel-wise QBN configurations having similar inference accuracy, latency and energy for MobileNetV2. We cannot differentiate these two configurations by using only the HAQ reward. However, the first configuration consumes $94\%$ of the FPGA area, while the other configuration occupies $85\%$ of the FPGA area. AutoQ can identify the second QBN configuration as a better choice via its extrinsic reward.

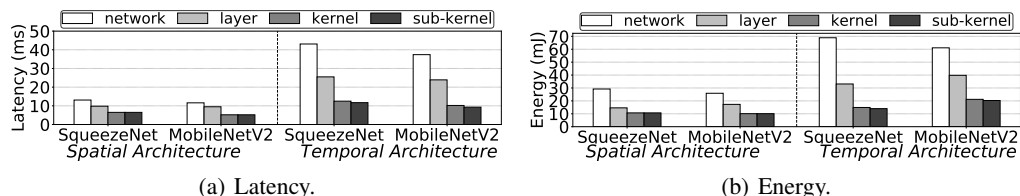

(a) Latency.  (b) Energy.

Figure 7: The comparison of latency and energy between temporal and spatial CNN accelerators.

**Quantization Granularity**. Besides the temporal CNN accelerator (Umuroglu et al., 2019b), the kernel-wise quantized models found by the accuracy-guaranteed AutoQ can reduce the inference latency on a spatial CNN accelerator, BitFunsion (Sharma et al., 2018), that relies on a 2D systolic array of the fusion units spatially summing the shifted partial products of weights and activations.As Figure 7 shows, compared to the layer-wise quantized models, on average, the kernel-wise quantized models reduce the inference latency by $39.04\%$ and decrease the inference energy by $33.34\%$ on the spatial CNN accelerator. Therefore, the kernel-wise quantized models greatly reduce the inference latency and energy on both the temporal and spatial CNN accelerators. Prior works (Mellempudi et al., 2017; Choukroun et al., 2019a) suggest it is possible to divide a weight kernel into several sub-kernels and quantize each sub-kernel independently. We also use AutoQ to search a QBN for each weight sub-kernel. As Figure 7 shows, the sub-kernel-wise quantized models cannot improve the inference latency or energy on the spatial CNN accelerator consisting of systolic computing arrays. Each dot-product operation of a sub-kernel-wise quantized model has to be split into several dot-product operations to be accumulated together. A systolic computing array still has to be designed to accommodate the weight sub-kernel with the largest QBN in a kernel. Therefore, we can see that it is difficult for the fine-grained quantization schemes choosing a QBN for each weight unit that is a

part of a kernel to further reduce the inference latency or energy on both the temporal and the spatial CNN accelerators.

## 5 CONCLUSION

In this paper, we propose a hierarchical-DRL-based kernel-wise network quantization technique, AutoQ, consisting of a HLC and a LLC. The HLC automatically searches an average weight QBN and an average activation QBN for each convolutional layer. Based on the average weight QBN, the LLC generates a QBN for each weight kernel in each layer. We also create a state space, a goal and action space, an intrinsic reward and an extrinsic reward to support AutoQ. Particularly, our shaped intrinsic reward enables the LLC to learn efficiently from the environment by considering both the HLC goal completion and the environment extrinsic reward. Moreover, the extrinsic reward of AutoQ can balance the inference accuracy, latency, energy consumption and FPGA area. Compared to the models quantized by the state-of-the-art DRL-based schemes, on average, the same models quantized by AutoQ reduce the inference latency by 54.06%, and decrease the inference energy consumption by 50.69%, while achieving the same inference accuracy.

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
