# OpenReview forum: "AutoQ: Automated Kernel-Wise Neural Network Quantization "
_ICLR.cc/2020/Conference — Accept (Poster)_

### Official Review · AnonReviewer1 · 2019-10-21
**Official Blind Review #1**

**Rating:** 3

**Review:**

This paper proposes a new method for quantizing neural network weights and activations that uses deep reinforcement learning to select the appropriate bitwidth for individual kernels in each layer. The algorithm uses a reward function that weights accuracy of the quantized model with latency, energy, and FPGA area, and leverages a high level and low-level controller to create quantized models that can take into account these factors. Compared to prior approaches taht only perform layer-wise instead of kernel-wise quantization, the quantized models can achieve better performance, or latency.

While the problem of more effectively quantizing neural network weights and activations is interesting, I found this paper hard to follow and evaluate. The proposed hierarchical deep RL method mentions a number of tricks and design decisions that are not ablated, and the notation and text around the method were difficult to follow. There were also no baselines comparing to other approaches for performing kernel-wise quantization (e.g. random search, quantize based off variance, etc.). Without improved baselines and text I would recommend rejecting this paper.

Major comments:
* Based on the analysis of kernel-wise results, it seems that a very simple strategy that chooses QBNs based on weight variance could be sufficient to achieve good performance. However, there’s no comparisons to these kinds of heuristics or even to random search over QBN per-kernel. There are also no ablations that study whether the hierarchical-RL based approach beats a baseline that just directly predicts the kernel-wise QBNs independently.  Without these baselines, it’s hard to know how well AutoQ works.
* The text that details the hierarchical RL approach with multiple controllers that is core to the new AutoQ approach is extremely hard to follow, with many undefined symbols.

Minor comments:
* FIg 1: where in the neightwork do these weight kernels come from? What network/dataset?
* You repeatedly state that ML experts obtain only sub-optimal results at selecting QBNs, can you cite something for this? What if you give experts access to additiional information like visualizations of weight distributions?
* What’s the tradeoff between using different QBNs for each kernel and specifying this number? Is there additional memory overhead?
* Table 1: hard to read the exponent in kernel-wise math, add parentheses
* Please use \citep vs. \citet where appropriate
* Why not also search for kernel-specific activation quantization?
* The notation in Fig 3 and surrounding text is really hard to follow, e.g. w_w, h_w, and indices into states (roundup could be ceil, names are confusing iRd, eRd?).
* Eqn 2 comes out of nowhere… why optimize for log(accuracy) vs accuracy? How do you choose the user-defined constants? Decay factor?
* Hardware overhead estimator: are these models accurate independent of QBN?
* Why do you have to perform Gaussian augmentation? I didn’t follow the re-labeling of transitions after Eqn. 6, is this needed?
* Define \mu when you first use it (Eqn 4?)
* Is \delta_a in implementation details the same as \sigma around Eqn 4?
* Table 3: Is there variance in these results based off fine-tuning? Could you include error bars or stderr on these estimates?
* Figure 6: are the search spaces different for these different approaches? I.e. does AutoQ perform better due to the upper/lower bounds you set on QBN?

=============

Thank you to the authors for addressing many of my concerns. The updated paper still does not present any baselines for methods that can search or learn kernel-wise quantization parameters, e.g. run random search and select the best-performing models according to your reward function. Additionally, there were no revisions made to the text to improve the presentation, and thus I continue to recommend rejection of this work.

**Experience Assessment:**

I do not know much about this area.

**Review Assessment: Checking Correctness Of Derivations And Theory:**

I assessed the sensibility of the derivations and theory.

**Review Assessment: Checking Correctness Of Experiments:**

I assessed the sensibility of the experiments.

**Review Assessment: Thoroughness In Paper Reading:**

N/A

---

> ### Author Response · Authors · 2019-11-10
> **Reply to reviewer 1**
>
> We would like to thank reviewer 1 for the thoughtful comments and efforts towards improving our manuscript.
>
> 1. Comparing against choosing QBNs based on weight variance
>
> First, simple heuristics like choosing QBNs based on weight variances cannot compete with our AutoQ, because of Equation 2, the Extrinsic Reward of AutoQ. AutoQ can not only do kernel-wise quantization but also directly consider the inference latency, hardware overhead and inference energy during the searches for QBNs. For example, in Table 3, we can see that different search policies such as resource-constrained and accuracy-guaranteed searches find totally different results. Although for ResNet-18, two search policies achieve only 2.4\% top-1 accuracy difference and 0.18\% top-5 accuracy difference, the inference latency difference is huge, i.e., >128\%. The simple heuristics cannot distinguish the latency, energy and hardware overhead difference at all.
>
> Second, although it is reasonable to choose a larger QBN (+A-bit) to quantize a weight kernel with larger variance (+B\%), it is difficult to decide how large QBN should be assigned to the kernel, i.e., the quantitative ratio between A and B. We tried such simple heuristic by setting a fixed ratio between A and B (e.g., 1:1 or 1:10), but the kernel-wise quantization result, i.e., accuracy, is much worse than HAQ [Wang et al. (2019)] and ReLeQ [Elthakeb et al. (2018)]. This is why we did not show such simple and naïve heuristics in the paper. If Reviewer-1 knows some papers show such simple heuristics can work, we are happy to read those references and add them as our strong baselines.
>
>
> 2. Fig 1: weight kernels? network/dataset?
>
> The weight kernels are from ResNet-18 on ImageNet.
>
>
> 3. Reference of ML experts obtaining only sub-optimal results?
>
> [Wang et al. (2019)] shows sometimes ML experts obtain only sub-optimal results during designing QBNs. ML experts are rare resources. AutoQ can automatically perform kernel-wise quantization for average users.
>
>
> 4. Extra memory overhead?
>
> We described the Storage Cost at the end of Section 3.
>
>
> 5. Kernel-specific activation quantization?
>
> Each activation layer of a CNN is defined as $\mathbf{A}\in \mathbb{R}^{n_{layer}\times c_{in}\times w_{a}\times h_{a}}$, where $n_{layer}$ is the number of layers; $c_{in}$ is the input channel number, $w_{a}$ is the feature map width, and $h_{a}$ means the feature map height. An activation layer has no kernel domain.
>
>
> 6. The notation in Fig 3.
>
> The notations in Figure 3 are consistent with the notations in Equation 1 – 6. $w_{w}$ indicates the weight kernel width, and $h_{w}$ is the weight kernel height. iRd is the Intrinsic Reward, while eRd is the extrinsic Reward.
>
>
> 7. In Eqn 2, log(accuracy) vs accuracy?
>
> Equation 2 is the most important equation in this paper. We define it as the Extrinsic Reward. Compared to directly using accuracy, log(accuracy) can make the HLC and LLC more sensitive to the accuracy degradation, so that they can find the QBN configurations with higher inference accuracy. User-defined constants are NOT decay factors. They are factor weights. Some design trade-offs can be decided by only users. For instance, AutoQ cannot decide the power consumption reduction is more important or the chip area reduction is more important for users. Only user themselves can decide which one is more important by assigning a larger weight to it.
>
>
> 8. Hardware overhead estimator accurate independent of QBN?
>
> Yes, the hardware overhead [Liu & Carloni (2013)] and power [Zhou et al. (2019)] estimators are trained by synthesis result mappings between QBN, hardware overhead and power consumption. It can accurately predict the hardware overhead and power consumption for each input QBN configuration.
>
>
> 9. Gaussian augmentation? Re-labeling transitions after Eqn. 6?
>
> AutoQ needs to perform the re-labeling. Because the LLC is learning and thus changing. The state-goal-reward transitions stored in the replay buffer are not 100% correct. These transitions obtained from the past LLCs do not accurately reflect the actions that would occur if the same goal was used with the current LLC.
>
>
> 10. $\mu$ in Eqn 4?
>
> $\mu_{\phi_{LLC}}^{LLC}(state_{[L_{i}, K_{j+1}]}$ is the policy network of LLC.
>
>
> 11. Is $\delta_a$ the same as $\sigma$ around Eqn 4?
>
> No, $\delta_{a_{[L_i,K_j]}}=0.5$ in the implementation details is the probability of LLC to perform explorations on its action during the search. In contrast, $\sigma_{a_{[L_i,K_j]}}$ in Equation 4 is the variance of the action during the exploitation.
>
>
> 12. Table 3: Variance in results?
>
> Error bars will be in the next version.
>
>
> 13. Figure 6: are search spaces different for different approaches?
>
> In Figure 6, all techniques use the same quantization method LQ-Nets and the same upper/lower bounds on the QBN during the search for MobileNetV2 on ImageNet. AutoQ outperforms because of its Hierarchical DRL agents with shaped intrinsic reward.

---

### Official Review · AnonReviewer3 · 2019-10-23
**Official Blind Review #3**

**Rating:** 8

**Review:**


Summary of the work

This paper proposes to automatically search quantization scheme for each kernel in the neural network. Importantly, due to a large amount of search space, hierarchical RL was used to guide the search.

Strength

First of all, I really liked the idea about more detailed search for kernel-wise configurations. It is interesting to see that different kernel settings has different best bitwidth. More importantly, it is great to provide cost model based estimations for cost/latency.

The authors also provided detailed discussion about weight bit distributions and average quantization bits among the layers.

The author provided implementation on real world FPGAs using bit-serial spatial architecture, which plays particularly nice with the algorithm. This something that should be highlighted in the discussion.

Weakness

First of all, I would love to see more detailed discussion about the corresponding hardware support implications. In particular, given the need of re-using computing resources, kernel-wise quantization has to use the bit-serial version of architecture(otherwise the MAC cannot be reused and we have to layout the entire network on FPGA), which may limit the applicability of the methods to higher number of bits.

The second potential weakness is the close relation between the method and HAQ. The method feels like a straight-forward extension(with DRL added). What would happen if you directly apply HAQ’s method to the kernel-wise search space?

Finally it would be great if the authors can clarify more about the FPGA setups(number of accumulators being used, whether there is reuse of compute unit in FPGA, or did you just layout everything spatially).

Question:

How do you handle layers like BatchNorm(which normally need floating pt)?

Overall, I find this paper provides interesting insights and solid evaluation and should be accepted to ICLR.


---
i have read the authors' response.

Please do note that because the different kernel-wise precision, the accelerator has to resort to bit-serial computation, which somewhat limits the structure of the accelerator. e.g. we cannot simply build 8bit MAC along with 4 bit ones, but have to use 4bit ones to bit-serially accumulate the 8 bit ones. This will somewhat limit the applicability of the model, and i think the author should add a discussion section to the paper about this limitation.


**Experience Assessment:**

I have read many papers in this area.

**Review Assessment: Checking Correctness Of Derivations And Theory:**

I assessed the sensibility of the derivations and theory.

**Review Assessment: Checking Correctness Of Experiments:**

I carefully checked the experiments.

**Review Assessment: Thoroughness In Paper Reading:**

I read the paper thoroughly.

---

> ### Author Response · Authors · 2019-11-10
> **Reply to reviewer 3**
>
> We would like to thank the reviewer 3 for the thoughtful comments and efforts towards improving our manuscript.
>
> 1. More details on hardware support. The kernel-wise quantization has to use the bit-serial architecture, which may limit the applicability of the methods to higher number of bits.
>
> The kernel-wise quantization uses bit-serial multiply-accumulation (MAC) units in both the temporal [Umuroglu et al. (2019b)] and the spatial [Sharma et al. (2018)] CNN accelerators. In each cycle, a bit-serial MAC unit can compute only 1-bit of a fixed-point MAC operation. For instance, if we quantize a kernel with 4-bit, a bit-serial MAC unit requires 4 cycles to complete a 4-bit MAC operation. However, compared to the conventional N-bit-parallel MAC unit, a bit-serial MAC unit costs only <1/N hardware overhead and <1/N power consumption. The CNN inference latency is decided by the accelerator MAC throughput. Although a bit-serial MAC unit increases the latency of an N-bit MAC operation by N times, both the temporal and the spatial CNN accelerators implement N times bit-serial MAC units to maintain the same MAC throughput with even smaller hardware overhead and power consumption. For higher number of bits, these bit-serial architectures can still have the same MAC throughput. Our kernel-wise quantization greatly reduces the average quantization bit number (QBN) of a CNN. Therefore, it can significantly increase the inference throughput of CNNs on both the temporal and the spatial CNN accelerators.
>
>
> 2. Directly applying HAQ’s method to the kernel-wise search space?
>
> Compared to the layer-wise quantization technique like HAQ, the search space of our kernel-wise quantization AutoQ is much larger as Table 1 shows. Therefore, it is difficult for traditional DRL DDPG-based agents to find optimal results for our kernel-wise quantization AutoQ. In Figure 6, compared to a DRL DDPG-based agent (HAQ), our hierarchical DRL with shaped intrinsic reward (AutoQ) uses shorter search latency to find a kernel-wise quantized CNN configuration with much higher inference accuracy.
>
>
> 3. Clarify the FPGA setups.
>
> The detailed implementation and configuration of the temporal accelerator is shown in [Umuroglu et al. (2019b)], while those of the spatial one can be found in [Sharma et al. (2018)]. The spatial CNN BitFunsion accelerator adopt a 2D systolic array of the fusion MAC units spatially summing the shifted partial products of weights and activations. The systolic array can automatically and spatially control and process the MAC data flow. We do not need to layout everything spatially.
>
>
> 4.	How do you handle layers like Batch Norm (which normally need floating pt)?
>
> We use fixed-point batch normalization during the kernel-wise quantized CNN inferences. Compared to its floating-point counterpart, the fixed-point batch normalization has little inference accuracy loss [Chen et al. (2017), Lin et al. (2016)].
>
>
> X. Chen, X. Hu, H. Zhou and N. Xu, "FxpNet: Training a deep convolutional neural network in fixed-point representation," 2017 International Joint Conference on Neural Networks (IJCNN), Anchorage, AK, 2017, pp. 2494-2501.
> Lin, Darryl, Sachin Talathi, and Sreekanth Annapureddy. "Fixed point quantization of deep convolutional networks." International Conference on Machine Learning. 2016.

---

### Official Review · AnonReviewer2 · 2019-10-24
**Official Blind Review #2**

**Rating:** 6

**Review:**

The paper proposed a method for network quantization. Similar with the work of "HAQ: Hardware-Aware Automated Quantization with Mixed Precision"(CVPR 2019), the proposed method is based on reinforcement learning. The contribution of the work is on the kernel-wise quantization, i.e., assigning different bitwidth to different kernels in one layer. And in the experiments, the proposed method clearly outperformed the state-of-arts of network-wise and layer-wise quantization methods.
Although the high level idea is presented very well, some essential parts of the paper is a little bit hard to follow.  The motivation of using the hierarchical DRL is unclear.

Questions:

Why a hierarchical DRL agent is desired for kernel-wise quantization? Is it possible to modify the definitions of state and action of HAQ for kernel-wise quantization, which seems to be a much simpler solution for the task?


What is the definition of iRd [L_i,K_j] in the Intrinsic Reward?


It seems the original HAQ used simple quantization way instead of LQ-Nets, which is different with the experiment setting in this paper. If I'm correct, does the change affect the HAQ's performance for fair comparison?




**Experience Assessment:**

I do not know much about this area.

**Review Assessment: Checking Correctness Of Derivations And Theory:**

I assessed the sensibility of the derivations and theory.

**Review Assessment: Checking Correctness Of Experiments:**

I assessed the sensibility of the experiments.

**Review Assessment: Thoroughness In Paper Reading:**

I read the paper at least twice and used my best judgement in assessing the paper.

---

> ### Author Response · Authors · 2019-11-10
> **Reply to reviewer 2**
>
> We would like to thank reviewer 2 for the thoughtful comments and efforts towards improving our manuscript.
>
> 1. The motivation of using hierarchical DRL.
>
> Compared to the layer-wise quantization technique like HAQ, the search space of our kernel-wise quantization AutoQ is much larger as Table 1 shows. Therefore, it is difficult for traditional DRL DDPG-based agents to find optimal results for our kernel-wise quantization AutoQ. In Figure 6, compared to a DRL DDPG-based agent (HAQ), our hierarchical DRL with shaped intrinsic reward (AutoQ) uses shorter search latency to find a kernel-wise quantized CNN configuration with much higher inference accuracy.
>
>
> 2. The definition of $iRd_{[L_i, K_j]}$ in the Intrinsic Reward.
>
> $iRd_{[L_i, K_j]}$ is the intrinsic reward for the weight kernel $K_j$ of the layer $L_i$. And it is used to evaluate the LLC action for the weight kernel $K_j$ of the layer $L_i$.
>
>
> 3. HAQ uses simple quantization. Why did we use LQ-Nets? Is it fair?
>
> To present the state-of-the-art inference accuracy, in this paper, we make both HAQ and AutoQ quantize CNNs by LQ-Nets in all figures and tables. It is a fair comparison. The only difference between HAQ and AutoQ in this paper is the quantization granularity, i.e., HAQ performs layer-wise quantization assigning a QBN to each layer, while AutoQ conducts kernel-wise quantization selecting a QBN for each kernel.

---

### Official Review · AnonReviewer4 · 2019-11-24
**Official Blind Review #4**

**Rating:** 6

**Review:**

Summary
This paper proposes a network quantization method. Different from previous methods focusing on network-level or layer-lever quantization, this work pays attention to kernel-level quantization. Specifically, they use a hierarchical reinforcement learning framework to search in the search space related with quantization. The experiment result validates the significance of the work.

Strength
The paper provides us with a new insight into network quantization. Even though the extension from layer-level to kernel-level is straightforward, the improvement is significant and meaningful. The experiment result demonstrates its efficiency in real applications.

Weakness
1. The algorithm of the paper is similar with previous work HAQ, which is based on DRL to guide the search procedure. Thus, the novelty of algorithm is somewhat weak.

2. For kernel-level quantization, this paper proposes a hierarchical DRL method. However, I didn't see the importance of the hierarchy. The author may discusses more about this and compare it with flatten algorithms for ablation study.

3. The paper didn't compare their methods with other baselines on the same level. I think some algorithms can be applied into kernel-level quantization directly. Based on the same level comparison, the efficiency of your method can be seen more directly.

4. The description was not written well. For example, the detail of the experiment and the hardware settings are unclear.


**Experience Assessment:**

I have read many papers in this area.

**Review Assessment: Checking Correctness Of Derivations And Theory:**

I assessed the sensibility of the derivations and theory.

**Review Assessment: Checking Correctness Of Experiments:**

I assessed the sensibility of the experiments.

**Review Assessment: Thoroughness In Paper Reading:**

I read the paper at least twice and used my best judgement in assessing the paper.

---

### Decision · Program_Chairs · 2019-12-19

**Decision:**

Accept (Poster)

**Comment:**

This paper proposes a network quantization method which is based on kernel-level quantization. The extension from layer-level to kernel-level is straightforward, and so the novelty is somewhat limited given its similarity with HAQ. Nevertheless, experimental results demonstrate its efficiency in real applications. The paper can be improved by clarifying some experimental details, and have further discussions on its relationship with HAQ.